# *Leishmania* Animal Models Used in Drug Discovery: A Systematic Review

**DOI:** 10.3390/ani13101650

**Published:** 2023-05-16

**Authors:** Jacob van der Ende, Henk D. F. H. Schallig

**Affiliations:** 1Fundación Quina Care Ecuador, Puerto el Carmen de Putumayo 210350, Sucumbíos, Ecuador; jacob.vanderende@quinacare.org; 2Experimental Parasitology Unit, Amsterdam University Medical Centres, Department of Medical Microbiology and Infection Prevention, Academic Medical Centre at the University of Amsterdam, 1105 AZ Amsterdam, The Netherlands; 3Amsterdam Institute for Infection and Immunity, Meibergdreef 9, 1105 AZ Amsterdam, The Netherlands

**Keywords:** *Leishmania*, leishmaniasis, animal studies, drugs

## Abstract

**Simple Summary:**

Animals are frequently used to determine whether new compounds have the potential to be used as drugs against the parasitic disease leishmaniasis. There is, however, no consensus on the best animal model, and many different ones are used. This review explored the many different models and identified their strengths and weaknesses. The work showed that many papers on animal models for leishmaniasis lack a thorough description of the experimental procedures used, are deficient in appropriate ethical review (to safeguard the welfare of the animals), and measures to reduce, refine or replace animal studies are hardly addressed. Mice and hamsters are most frequently used, but there are many differences in how the animal experiment is set up. We propose a more generalized animal model, in particular to ensure that different experiments can be better compared. Furthermore, we make a plea to report all animal experiments in the same way, and all experiments must be scrutinized by appropriate ethical review committees.

**Abstract:**

Many different animal models are in use for drug development for leishmaniasis, but a universal model does not exist. There is a plethora of models, and this review assesses their design, quality, and limitations, including the attention paid to animal welfare in the study design and execution. A systematic review was performed following the Preferred Reporting Items for Systematic Reviews and Meta-analyses (PRISMA) guidelines of available literature after the year 2000 describing animal models for leishmaniasis. The risk of bias was determined using the SYstematic Review Centre for Laboratory animal Experimentation (SYRCLE) risk of bias assessment tool. A total of 10,980 records were initially identified after searching the databases PubMed, EMBASE, LILACS, and SciELO. Based on the application of predetermined exclusion and inclusion criteria, a total of 203 papers describing 216 animal experiments were available for full analysis. Major reasons for exclusion were a lack of essential study information or appropriate ethical review and approval. Mice (82.8%; an average of 35.9 animals per study) and hamsters (17.1%; an average of 7.4 animals per study) were the most frequently used animals, mostly commercially sourced, in the included studies. All studies lacked a formal sample size analysis. The promastigote stages of *L. amazonensis* or *L. major* were most frequently used to establish experimental infections (single inoculum). Animal welfare was poorly addressed in all included studies, as the definition of a human end-point or consideration of the 3Rs (Replacement, Reduction, Refinement) was hardly addressed. Most animals were euthanized at the termination of the experiment. The majority of the studies had an unknown or high risk of bias. Animal experiments for drug development for leishmaniasis mainly poorly designed and of low quality, lack appropriate ethical review, and are deficient in essential information needed to replicate and interpret the study. Importantly, aspects of animal welfare are hardly considered. This underpins the need to better consider and record the details of the study design and animal welfare.

## 1. Introduction

Leishmaniasis is a protozoan parasitic disease caused by *Leishmania* species [1,2]. The parasites are transmitted through the bite of an infected female sand fly (*Phlebotominae*), and there are about 20 *Leishmania* species that can infect humans [1,2]. Leishmaniasis is frequently encountered in (sub-)tropical regions. Depending on the species, a variety of clinical symptoms can occur, ranging from (large) skin ulcers (cutaneous leishmaniasis) to the destruction of mucosal membranes (of nose and mouth; muco-cutaneous leishmaniasis) to the lethal invasion of internal organs (visceral leishmaniasis) [1,2,3]. Next to a severe impact on health, leishmaniasis can have massive economic and social consequences [4], including the loss of quality of life and (severe) the stigmatization of affected people [5,6]. Leishmaniasis is classified as one of the most important neglected tropical diseases, affecting the poorest of the poor, and is associated with malnutrition, poor housing, people displacement, and a lack of financial resources [1,4]. 

The treatment of leishmaniasis is difficult and painful, and many drugs have significant toxic side effects [1,2,3]. Many of the available and accessible drugs are old and require repeated administration. In addition, there is emerging drug resistance [7,8]. A registered vaccine against human disease is not yet available [9,10]. Therefore, there is a continuous need to develop new medicines against leishmaniasis [11,12]. However, there are hardly any drugs under development for leishmaniasis due to limited interest from pharmaceutical companies. This is due, amongst other reasons, to the chances for return-of-investment for these investments being very low. Fortunately, public–private partnerships, such as the drugs for neglected diseases initiative (DNDi), work towards the delivery of new treatment options.

A key step in the development of new drugs for leishmaniasis remains the pre-clinical assessment of the safety and efficacy of new components in animal models [13]. There are currently many different animal models in use, and these in vivo models for the discovery of new potential anti-leishmanial drugs differ hugely in many aspects [14]. There is a vast variation in animals used (species, strain, sex), *Leishmania* species used for experimental infection, the dose and route of the inoculation of the challenge strains, responses to the disease, and the clinical presentation of symptoms [13,14]. This disparity in models makes it difficult to draw conclusions and to make comparisons between different in vivo results. 

This review aims to assess the design, quality, and limitations of various animal models for leishmaniasis their use in drug discovery and evaluation. It further focuses on the attention that is being paid to animal welfare in study design and execution.

## 2. Methods

### 2.1. Search Methodology

This systematic review followed the Preferred Reporting Items for Systematic Reviews and Meta-analyses (PRISMA) guidelines [15]. A systematic search was performed using the PubMed https://pubmed.ncbi.nlm.nih.gov (accessed on 29 March 2020), Embase https://www.embase.com (accessed on 29 March 2020), the Latin American and the Caribbean Health Sciences Literature database (LILACS); http://lilacs.bvsalud.org (accessed on 29 March 2020), and the Scientific Electronic Library Online (SciELO); https://www.scielo.br (accessed on 29 March 2020) databases.

Many animal models for leishmaniasis are mice (*Mus* spp.), rabbits (*Oryctolagus* spp.) or hamsters (*Cricetinae* spp.) [13,14]. For the purpose of this review, we did not restrict ourselves to these species, but did restrict the search to mammals to exclude animal species that are unsuitable to be used as a standard animal model for human leishmaniasis, such as fish species, birds or invertebrates. Specific search filters for increasing the retrieval of animal studies in Embase [16] and PubMed [17] were applied.

The databases were last searched on 29 March 2020. The complete search syntaxes are presented in Appendix A. Language restriction was applied, and the inclusion was restricted to manuscripts in English, Dutch, Afrikaans, Portuguese, and Spanish. Publications after 1 January 2000 were considered for inclusion.

Full papers presenting original data of which the full text could be retrieved were used for review. Excluded were publications of which the full text was not retrievable, editorials, conference abstracts, conference proceedings, case reports, and (systematic) reviews.

The retrieved records were imported into EndNote (EndNoteX9; https://endnote.com accessed on 22 March 2023), and replicates were removed using this software package followed by the further removal of replicates by using Rayyan web-based (https://rayyan.ai; accessed on 22 March 2023) software [18]. A further reduction of included papers was achieved by applying the language and publication date restrictions and removing non-primary papers using the Rayyan application.

### 2.2. Study Eligibility

After the removal of duplicates, the title and abstract of the articles were screened using pre-determined in- and exclusion criteria by two independent readers (JvdE and HDFHS).

A second screening was performed on full-text articles. Discrepant results were resolved by discussion until a unanimous decision was reached.

We included papers for review that reported in vivo animal experiments on the safety or efficacy of compounds for the treatment of leishmaniasis. Papers addressing vaccine development, studies on immune responses to *Leishmania*, in vitro studies, non-mammalian studies or papers that addressed infections other than leishmaniasis were excluded.

### 2.3. Data Extraction

For each eligible publication, the following study variables were extracted: whether there was an ethical review (and approval) of the study; species, breed and gender of animal used for experimentation; weight and/or age of the experimental animal; number of animals used in total and per group; source (commercial or in-house bred); housing (individual or per group); pre-treatment(s) given; use of pregnant animals; information on the experimental challenge: i.e., *Leishmania* species, stage, and number of parasites used for inoculation (promastigotes or amastigotes), route and frequency of inoculation, duration of inoculation; whether next to an experimental treatment a competitor treatment, placebo treatment, and/or no treatment was applied; duration to read-out, primary organs targeted for read-out and experimental method used for read-out (i.e., lesion size, microscopy, molecular biology, limiting dilution assay, or other methods); physiological parameters reported (i.e., weight and blood parameters); and statistical analysis performed (i.e., sample size calculation and statistical method used to analyze data). In addition, information on animal welfare (i.e., housing (individual or per group; access to food and water); consideration of a human end-point [19], consideration of Replacement, Reduction, Refinement (3Rs) [20], and what happened with the experimental animal at the end of the experiment were also extracted. Two reviewers performed the data extraction, and any disagreements were resolved by consensus.

A manuscript was immediately excluded from full analysis if an ethical review (and/or approval) of the study protocol was not performed or reported. Furthermore, a study was not further analyzed if one of the following primary parameters was missing (essential study information, which is needed at a minimum to be able to repeat the animal experiment [21]): species/breed, weight or age, number of animals used (total or per group), and information on experimental *Leishmania* infection.

### 2.4. Risk of Bias

The quality of the animal experiments described in papers included for full analysis was assessed with the SYRCLE´s risk of bias (RoB) assessment tool [22]. The RoB tool for animal studies contains 10 entries, which are related to selection bias, performance bias, detection bias, attrition bias, reporting bias, and other biases. Two reviewers performed the RoB assessment, and any disagreements were resolved by consensus.

### 2.5. Registration

This systematic review is registered in the international prospective register of systematic reviews (PROSPERO: https://www.crd.york.ac.uk/prospero/ accessed on 12 May 2023) under registration number CRD42019092382.

## 3. Results

### 3.1. Search Results

In total, 10,980 records were retrieved after searching the databases PubMed, EMBASE, LILACS, and SciELO. After the removal of replicates, the database comprised 7369 records. Subsequently, 1731 records were removed because they were published before 1 January 2000. Another 259 records were removed because they did not comply with the language restrictions or were not primary papers or focused on non-mammals. In total, 5379 records were selected for title and abstract reading. The screening of titles and abstracts resulted in the removal of another 4763 papers. The major reasons for exclusion were that the paper described a vaccine study or was focused on the immunology research of leishmaniasis infections. Other reasons were that the manuscript was a background or review paper or it did not address an in vivo study on leishmaniasis.

In total, 616 papers remained for full text assessment after the title and abstract selection. During the process of full text analysis, another 77 articles were removed from the analysis, mainly because either the full text could not be accessed or the paper turned out to be a conference abstract (without sufficient details to perform an analysis) or a review.

Finally, 539 articles, describing 634 animal experiments, were available for further analysis. From this set of articles, 216 papers were excluded because appropriate ethical review or approval was not reported. Furthermore, another 120 articles were excluded because essential study information was not reported. This finally resulted in 203 articles that remained for data extraction and full analysis. A full list of the references of the papers included in this systematic review is presented as Appendix A. Some of these papers reported more than one animal experiment. In total, 216 animal experiments were available for analysis (see Figure 1).

### 3.2. Experimental Animals

None of the included studies used pregnant animals. Pre-treatments, for example, steroids to modulate the immune response or to make the animals more susceptible to *Leishmania* infection, were not given to the animals in the analyzed experiments.

Mice (*Mus musculus*), and in particular the breed BALB/c (92.2%; 165/179), were by far the most frequently used experimental animals in the analyzed animal studies (Table 1). Swiss strain albino mice (3.4%%; 6/179) and the C57BL/6 strain (2.2%; 4/179) were most often used. CBA/J inbred mice were used in two studies (1.1%), and genetically engineered C3H/HePas or P2 × 7−/− mice were each used in one study (1.1%).

Female mice were used in 69.3% (124/179) of the animal studies with *M. musculus* breeds; 12.3% (22/179) of the studies used male mice, 2.2% (4/179) used both female and male mice in the experiment, and 16.2% (29/179) did not report the gender of the used mice. The mean age of the mice was 6.8 weeks (95% CI: 6.5–7.1), and the mean weight was 22.7 g (95% CI: 21.3–24.1).

The Syrian golden hamster (*Mesocricetus auratus*) was used in 37 analyzed studies (Table 1), of which 24.3% (9/37) used female specimens, 51.4% (19/37) male, 13.5% (5/37) male and female mixed, and 10.8% (4/37) did not report the gender of the used hamsters. The mean age of the hamsters was 8.8 weeks (95% CI: 6.8–10.8), and the mean weight was 90.8 g (95% CI: 76.2–105.3).

Rats and dogs were reported in very few initially included studies (*n* = 4), but these were not included in the final analysis because of a lack of essential study information.

### 3.3. Sample Size Calculations and Applied Statistics

None of the analyzed studies reported a formal sample size calculation. The mean number of mice per study group was 6.5 animals (95% CI: 5.6–7.4 animals per group)—see Table 1. The mean total number of mice per study was 35.9 (95% CI: 25.5–46.4 mice per study).

The mean number of hamsters per study group was 7.4 animals (95% CI: 6.9–7.9 animals per group)—see Table 1. The mean total number of hamsters used per study was 36.4 (95% CI: 32.9–39.8 animals per study).

In total, 20 studies (9.3%) did not report the application of a formal statistical analysis of the data generated by the animal experiment. All other studies (*n* = 196; 90.7%) did report on the statistical method that was applied. The most frequently used statistical analysis was ANOVA, followed by Student’s *t*-test or Tukey test.

### 3.4. Experimental Design

All studies reported the inclusion of a group of animals that was subjected to treatment with an experimental anti-leishmanial drug or compound (Table 2). A competitor treatment with an established anti-leishmanial drug (such as miltefosine, pentamidine or pentostam), to establish the safety or efficacy of the experimental treatment, was reported in 66.2% (143/216) of all included studies. A placebo treatment (a substance without any known therapeutical activity such as saline solutions) was included in 73.1% (158/216) of the analyzed experiments.

*L. amazonensis* was the most frequently used *Leishmania* species to establish an experimental infection in mice, followed by *L. major* (Table 2). *L. donovani* was most often used in hamster experiments. It was noted that of all main *Leishmania* species causing disease, *L guyanensis* was not used as inoculum in all included studies. The promastigote stages of *Leishmania* species were by far most frequently used to establish an experimental infection (84.3%; 182/216). In contrast, (axenic) amastigote stages were used in 34 experiments (15.7%).

In all studies, a single inoculum was given to establish an experimental *Leishmania* infection. None of the animals was repeatedly infected with the parasites. The mean number of parasites used to establish an experimental infection in mice was 1.25 × 10^7^ ± 2.31 × 10^7^. In general, the inoculum in hamsters was higher: 2.36 × 10^7^ ± 3.26 × 10^7^. A sub-cutaneous inoculation was the most frequently used method of administration (65.3%; 141/216) to establish the experimental infection (Table 2). The duration of inoculation (i.e., the time between the administration of the experimental infection and the (first) read-out to measure the effect of the administered drug or compound) was slightly longer than 4 weeks for both experimental species (Table 2).

The experimental techniques used as read-out to assess effect of the administered drug or compound are also presented in Table 2. The most frequently applied technique was the measurement of the lesion size or the increased thickness of the infected foot pad, followed by limiting dilution assay to determine number of *Leishmania* parasites in tissue samples [23]. Next, microscopy, either the determination of the number of Leishman–Donovan units [24] in tissue slides or histopathology, or molecular techniques were most frequently used. The skin was the most often targeted organ for read-out, followed by spleen/liver or lymphatic tissue (see Table 2).

Physiological parameters reported in the studies were weight gain or loss (16.2% of the analyzed studies) or blood parameters including cytokines, antibodies, and inflammatory markers (33.8%).

### 3.5. Animal Welfare Aspects

Animal welfare was poorly addressed in all included studies (Table 3). None of the studies reported the definition of a human end-point to terminate the experiment. There were 12 studies (5.6%) that reported consideration of the 3Rs to a certain extent. These considerations mostly reported not including an uninfected control group but instead used another footpad as a reference for the development of *Leishmania* infection-associated lesions. Most animals were sacrificed at the end of the experiment (Table 2).

The sourcing of experimental animals was reported in 69.4% (150/216) of the studies; 60.2% (130/216) of these studies used animals obtained from a commercial source, and 9.3% (20/216) used animals reared in-house or from another non-commercial source. However, sourcing was not reported in 30.6% (66/216) of the studies (Table 3).

Housing conditions were reported for 35 (16.2%) of the analyzed studies, but not for the majority (*n* = 181; 83.8%). Housing in groups was reported for 10 studies (4.6%), and individual housing was reported in 25 studies (11.6%). Furthermore, 91 studies (42.1%) provided information on access to food and water, lighting conditions, humidity or housing under spf conditions.

### 3.6. Risk of Bias Assessment

SYRCLE’s risk of bias tool for animal studies was applied for the evaluation of the quality of the included studies [22]. In total, 10 criteria were interpreted to assign a judgment of a low (if every detail of the criterion was described), a high (if information was not available) or an unknown (if there was insufficient information) risk of bias. 

After the evaluation of the criteria, it was observed that there was a significant risk for selection bias, in particular for the allocation procedures applied in the animal studies. For 30% of the studies, it was reported that some kind of randomization took place, but further details on how the allocation sequence was generated and applied were missing. All other studies (70%) did not report on randomization at all. Moreover, none of the studies (100%) reported whether the allocation was adequately concealed (high risk of selection bias). In contrast, with the exception of three studies, in most studies, the study groups were similar at baseline (low risk of selection bias). All analyzed studies had a high risk of performance bias as none reported on the random housing of the animals during the experiment and whether caregivers and/or investigators were blinded from knowledge on which intervention each animal received during the experiment. Further, in terms of detection bias, the risk was determined to be high as most studies failed to report whether the animals were selected at random for outcome assessment, and it was not reported whether the outcome assessor was blinded. For all studies, there was an unknown risk of attrition bias. None of the studies reported on incomplete outcome data or reported on having encountered these, but all presented complete data sets. Finally, it was determined that almost all studies (with the exception of two) were free of selective outcome reporting (low risk of bias) and free of other problems that could result in a high risk of bias. This was most likely due to the fact that the inclusion and exclusion criteria for this systematic review were stringent.

## 4. Discussion

This literature review presents a systematic analysis of the animal models that are in use to determine the efficacy of (experimental) drugs for leishmaniasis. Although there is a wealth of publications in this field, a critical review signals that many of these studies are poorly designed, lack essential information that is needed to reproduce experimental data, and have an unknown or high risk of bias. This is despite the availability of guidelines, i.e., the ARRIVE guidelines (Animal Research: Reporting of In Vivo Experiments) that were originally developed in 2010 to improve the reporting of animal research [25] and updated in 2020 [26]. Moreover, animal welfare requirements were hardly reported in the studies included in this review. Furthermore, attempts to achieve the 3Rs and to define human endpoints are scarcely considered whilst designing experiments [19,20]. A more stringent approach in evaluating protocols by scientific and ethical committees towards these points is therefore called upon. These observations further underpin the requirement that scientific journals publishing animal experiments need to emphasize to record details concerning animal welfare [21,27] and that at least an ethical review and approval of the study should be provided.

A multiplicity of experimental animal protocols is presented with a huge variation in the choice of animals (mainly mice and hamsters), all with different ages, weights and genders, inocula (including a large variety of different *Leishmania* species), duration of experiments, and read-outs. This makes it impossible to extract a complete generalized protocol or model to study (experimental) drug efficacy in animals from the analyzed studies. Nevertheless, having a standardized model available would greatly improve the comparability of different studies on the efficacy of anti-leishmanial (experimental) drugs and would contribute to the reduction in the number of animals used [11,12]. Therefore, we propose the following more generalized animal model based on this systematic review (extracted from Table 1 and Table 2).

To standardize animal experiments, we suggest the use of female BALB/c mice with an average age of 6 to 7 weeks and a weight of 22 ± 2 g. BALB/c mice are preferred over other strains as they are known to present a high susceptibility phenotype (and thus may display a more profound outcome for treatment outcome) as compared to, for example, Swiss mice that are highly resistant to *Leishmania* infections, in line with observed responses in humans [28].

The selection of infecting *Leishmania* species will strongly depend on what clinical type of leishmaniasis will be studied, but we propose that for visceral leishmaniasis (VL) drug studies, in the first instance, *L. donovani* should be used, compared with *L. major* for Old World cutaneous leishmaniasis (CL) and *L. amazonensis* for New World CL. Preferably, WHO reference strains of *Leishmania* will be used to further optimize the standardization of the model. Other species can be studied when sufficient efficacy has been established. We propose an initial infective dose of 1.25 × 10^7^ promastigotes and a duration of 21 days of infection to establish disease. The group size could be six to seven animals per experimental group, although a formal sample size analysis has not been done. The experiment should include at least one group that is treated with an established drug to treat leishmaniasis of interest and a placebo group. Most likely, a non-treated (as this is covered by the placebo) and a non-infected group (as this has no added value) are not needed. Several different read-out systems are used to determine the efficacy of the drug under investigation. Many of them are in-house developed and/or prone to subjective interpretation, such as microscopy. To circumvent this and to establish an objective quantifiable read-out system, we propose the use of standardized quantitative PCR methods for VL and CL [29,30,31]. To allow the determination of the *Leishmania* parasite burden over multiple time points in the same animal, the use of in vivo imaging using luciferase/fluorescent parasites could be used as a more advanced method [32,33]. However, this technology might be technically more demanding and requires the use of genetically modified *Leishmania* strains and extensive laboratory infrastructure. A multi-center validation of the proposed animal model, including read-out systems, should be performed.

The study protocol must always be ethically reviewed by a competent committee and must include sufficient information on the expected efficacy of the drug under experimental investigation that justifies the initiation of an in vivo animal experiment. Finally, aspects of animal welfare (i.e., housing, formulation of human end-points, consideration of 3Rs) should always be addressed in the protocol and reported in the scientific publication [19,20,25,26,27].

## 5. Conclusions

Animal experiments for the drug development for leishmaniasis are in mainly poorly designed and of low quality, lack appropriate ethical review, and are deficient in essential information needed to replicate and interpret the studies. Importantly, aspects of animal welfare are hardly considered. This underpins the need to better consider and record the details of the study design and animal welfare.

## Figures and Tables

**Figure 1 animals-13-01650-f001:**
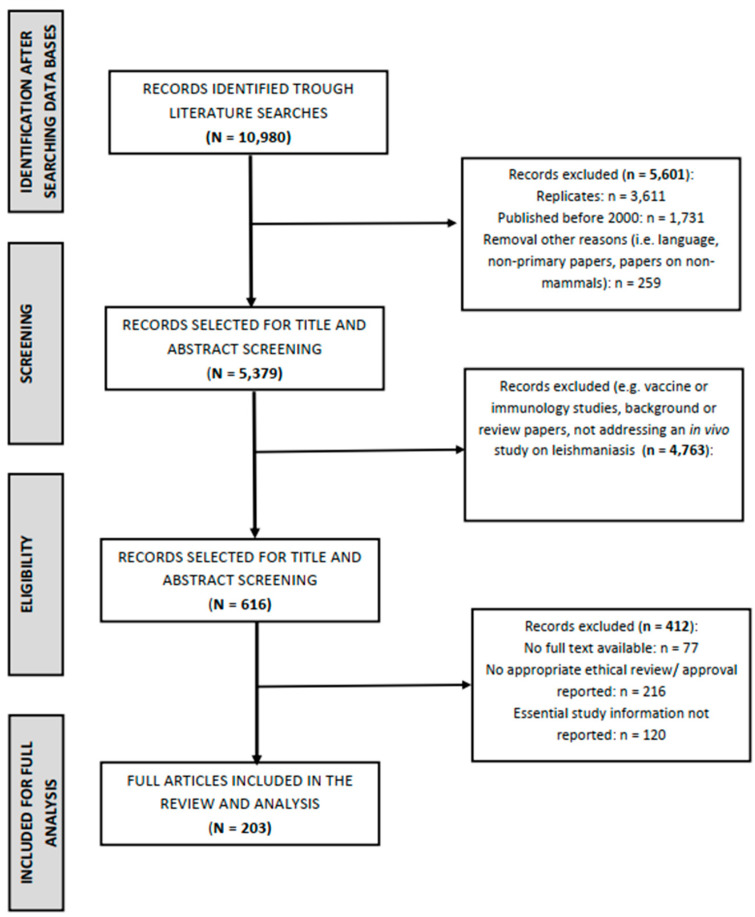
Results of the literature search and selection process. Following the exclusion of publications that did not meet the primary inclusion criteria, a total of 616 eligible studies were finally included for full text review. After application of the criteria for ethical review and approval or the availability of essential information for the conduct of the animal experiment (i.e., species/breed, weight or age, number of animals used (total or per group), and information on experimental *Leishmania* infection), 204 articles describing 216 animal studies remained for full analysis.

**Table 1 animals-13-01650-t001:** General information on the experimental animals. Information on gender, mean age (in weeks ± standard deviation and 95% confidence interval [95% CI]), mean weight (in grams ± standard deviation and 95% confidence interval [95% CI]), and mean number in total per study (±standard deviation and 95% confidence interval [95% CI]) and per group (±standard deviation and 95% confidence interval [95% CI]) of the experimental animals in 216 analyzed studies.

	Mouse (*Mus musculus*) *; 179 Studies	Hamster (*Mesocricetus auratus*); 37 Studies
Gender	124 female; 22 male: 4 female and male mixed; 29 gender unreported.	9 female; 19 male: 5 female and male mixed; 4 gender unreported.
Mean age (weeks)	6.8 ± 0.2 [6.5–7.1]	8.8 ± 1.0 [6.8–10.8]
Mean weight (grams)	22.7 ± 0.7 [21.3–24.1]	90.8 ± 7.1 [76.2–105.3]
Mean number in total per study	35.9 ± 5.1 [25.5–46.4]	36.4 ± 1.7 [32.9–39.8]
Mean number in total per study group	6.5 ± 0.4 [5.6–7.4]	7.4 ± 0.3 [6.9–7.9]

* All breeds used combined.

**Table 2 animals-13-01650-t002:** Experimental study design. Information on the design of the animal experiments (*n* = 216): treatment (experimental, competitor or placebo) given to experimental groups; *Leishmania* species used to induce experimental challenge infections; stage of parasite used for experimental infection; mean number of parasites used to establish experimental infections (±standard deviation); route of inoculation to establish challenge infection; mean duration of inoculation in days (±standard deviation) before the first read-out of the effect on experimental challenge infections was performed; experimental techniques used as read-out; reported status of the experimental animals at the end of the experiment.

	Mouse (*Mus musculus*) ^1^; 179 Studies	Hamster (*Mesocricetus auratus*); 37 Studies
**Treatment provided**		
Experimental treatment ^2^	179 (100%)	37 (100%)
Competitor treatment ^3^	115 (64.2%)	28 (75.7%)
Placebo treatment ^4^	133 (74.3)	25 (67.6%)
***Leishmania* species**		
*L. amazonensis*	67	4
*L. major*	48	1
*L. donovani*	29	12
*L. infantum*	24	6
*L. chagasi* ^5^	1	3
*L. brazaliensis*	6	8
*L. mexicana*	2	-
*L. tropica*	2	1
*L. panamensis*	-	2
**Stage of *Leishmania*** used for inoculation		
Amastigote	21	13
Promastigote	158	24
**Mean number (log) of parasites per inoculum** to establish experimental infection	1.25 × 10^7^ ± 2.31 × 10^7^	2.36 × 10^7^ ± 3.26 × 10^7^
**Route of inoculation**		
Intravenous	30	-
Intracardial	7	14
Intraperitoneal	15	7
Subcutaneous	125	16
Other	2	-
**Mean duration of inoculation** (in days) ± standard and 95% CI	22.9 ± 1.6 [26.8–33.0]	31.4 ± 3.3 [24.7–38.1]
**Experimental technique used as read-out to assess the effect of treatment ^6^**		
Lesion size or increased thickness of the foot pad	109	15
Microscopy (Leishman–Donovan units)	28	6
Microscopy (Histopathology)	20	12
Limiting dilution assay	80	12
(q)PCR	31	2
Other methods	18	2
**Primary organ targeted for read-out ^7^**		
Skin (foot pad)	57	13
Skin (base of tail)	36	0
Skin (other location)	7	3
Spleen and/or liver	77	21
Bone marrow or other lymphatic tissue	36	9
Other organs	21	3
**Other physiological parameters reported**		
Weight (gain or loss)	30	5
Blood parameters (inc. cytokines, antibodies, inflammatory markers)	67	6
**Reported status of the experimental animals at the end of the experiment**		
Death	162	32
Alive	3	1
Not reported/unclear	19	4

^1^ All breeds of mice used combined; ^2^ Experimental treatment is a treatment (or a combination of) given to experimental animals of which the efficacy or safety was not established previously; ^3^ Competitor treatment is a treatment with an established anti-leishmanial drug for human or veterinary use (for example, miltefosine, pentostam or pentamidine etc.); ^4^ Placebo treatment is a treatment with either a vehicle (substance used to administer the experimental treatment) or a substance without therapeutic value (such as saline and ethanol solutions without any other active component); ^5^
*L. chagasi* is nowadays considered to be the same species as *L. infantum*, but it was reported as a separate species in these four studies; ^6^ Some studies used more than one technique for read-out; ^7^ Some studies targeted more than one organ/tissue for read-out.

**Table 3 animals-13-01650-t003:** Animal welfare aspects. Information on sourcing, housing, the definition of human endpoints to terminate study participation, consideration of the 3Rs (i.e., replacement, reduction, refinement) for animal experiments, and the reporting of other animal welfare issues (access to food and water, humidity, lighting conditions and housing under specific pathogen-free conditions) for the 216 analyzed animal experiments.

	Mouse (*Mus musculus*) *; 179 Studies	Hamster (*Mesocricetus auratus*); 37 Studies
**Sourcing**		
Commercial breeder	108 (60.3%)	22 (59.5%)
In-house, non-commercial breeder	17 (9.5%)	3 (8.1%)
Not reported	54 (30.2%)	12 (32.4%)
**Housing**		
Individual	21 (11.7%)	4 (10.8%)
In groups	6 (3.4%)	4 (10.8%)
Not reported	152 (84.9%)	29 (78.4%)
**Definition of human endpoint reported**	0 (0.0%)	0 (0.0%)
**3Rs considered**	10 (5.6%)	2 (5.4%)
**Other welfare issues reported**	77 (43.0%)	14 (37.8%)

* All breeds used combined.

## Data Availability

The data used to support the findings of this study are available from the corresponding author upon request.

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
