# Peer review of "Leishmania Animal Models Used in Drug Discovery: A Systematic Review"

_animals, 2023, doi:10.3390/ani13101650_

Round 1
Reviewer 1 Report
The article is novel, and there is a need for a systematic review of animal models used in leishmaniasis. They review the animal model used for cutaneous (CL), mucocutaneous (MCL) and visceral leishmaniasis (VL), which is novel. However, a major revision is needed to be accepted. The authors can find some comments in the following.
1-Please edit the title. (experimental) should be omitted. they can use drug discovery or antileishmanial drugs in the title,
2-In the methodology, they did not include all the animal models.
3- Fig 1 should be edited.
4-The authors mentioned the animal models used for all the species, so they should classify animal models based on the clinical forms and species and discuss them in detail. It would be better to address the difference between clinical forms in animal models in the Tables.
5-They should include articles which used other species, such as L. guyanensis, and L. aethiopica, in animal models.
6-Piramates also used an animal model in leishmaniasis, so please mention them in the tables and text.
7-They did not divide the Rodent models in Table 2; It would be better to specify the strains, such as BALB/c, C57BL/6, CBA, and CsS-16, based on the sensitivity to Leishmania spp.
8- Each of the animal models, including mouse models, rat models, hamsters, dog, and non-human primate models, should be addressed in separate paragraphs and also in the Tables and discussed based on the variables.
9-Tables 1, 2 and 3 should be edited, and other animal models should be added.
10-The Yucatan deer mouse (Peromyscus yucatanicus) was not mentioned as an animal model. Please state in the text and Tables.
Author Response
COMMENTS REVIEWER 1
The article is novel, and there is a need for a systematic review of animal models used in leishmaniasis. They review the animal model used for cutaneous (CL), mucocutaneous (MCL) and visceral leishmaniasis (VL), which is novel. However, a major revision is needed to be accepted. The authors can find some comments in the following.
1-Please edit the title. (experimental) should be omitted. they can use drug discovery or antileishmanial drugs in the title,
R: Title has been changed according to the suggestion. The title is now as follows: Animal models for drugs discovery for leishmaniasis: a systematic review
2-In the methodology, they did not include all the animal models.
R: We made a general comment that mostly mice or hamsters are being used, but for the purpose of this review we included in our search strategy in principle all mammals. Unfortunately no other animals could finally be included in the SR as papers reporting on these did not meet our inclusion criteria.
3- Fig 1 should be edited.
R: We ask the editorial office to edit this. We have submitted a correct figure, but can not edit ourselves.
4-The authors mentioned the animal models used for all the species, so they should classify animal models based on the clinical forms and species and discuss them in detail. It would be better to address the difference between clinical forms in animal models in the Tables.
R: Only two models were in the end of the day included in the review, mice and hamsters, as all other failed to meet our inclusion criteria. We have presented these two models separately throughout the manuscript.
5-They should include articles which used other species, such as L. guyanensis, and L. aethiopica, in animal models.
R: Our search strategies included these species, but there were no papers that meet our inclusion criteria for the SR that were focussing on these species. Consequently we are not able to report on L. guyanensis, and L. aethiopica.
6-Piramates also used an animal model in leishmaniasis, so please mention them in the tables and text.
R: Our search strategies included these species, but there were no papers that meet our inclusion criteria for the SR that were focussing on these species.
7-They did not divide the Rodent models in Table 2; It would be better to specify the strains, such as BALB/c, C57BL/6, CBA, and CsS-16, based on the sensitivity to Leishmania spp.
R: Most mice models use Balb/C, the other strains are not much used. Therefore, we prefer to keep them as one group.
8- Each of the animal models, including mouse models, rat models, hamsters, dog, and non-human primate models, should be addressed in separate paragraphs and also in the Tables and discussed based on the variables.
R: Only mice and hamster models were included in this review. Other animal models did not meet our inclusion criteria and are thus not reported (see lines 204-205: Rat and dog was reported in very few initially included studies (n= 4), but these were not included in the final analysis, because of lack of essential study information. Hamster and Mice models are separately presented in the manuscript.
9-Tables 1, 2 and 3 should be edited, and other animal models should be added.
R: Only mice and hamster models were included in this review. Other animal models did not meet our inclusion criteria and are thus not reported. Hamster and Mice models are separately presented.
10-The Yucatan deer mouse (Peromyscus yucatanicus) was not mentioned as an animal model. Please state in the text and Tables.
R: Indeed there are a few studies reporting on the use of this animal as a model. See for example: Loría-Cervera EN, Sosa-Bibiano EI, Van Wynsberghe NR, Andrade-Narváez FJ. Finding a model for the study of Leishmania (Leishmania) mexicana infection: The Yucatan Deer mouse (Peromyscus yucatanicus) as a suitable option. Acta Trop. 2018 Nov;187:158-164. doi: 10.1016/j.actatropica.2018.08.003. However, the available studies do not report on its use as a model to study novel therapeutics and has consequently not been included.
Reviewer 2 Report
This is an interesting manuscript and it reviews an important area. I have the following suggestions the authors may like to consider.
1. Line 62. I suggest the authors amend the sentence to remove the word dangerous and this is misleading. Clinicians do not administer dangerous levels of drugs – they treat with a prescribed dosage based on the person’s weight and other factors. The problem with many antileishmanial drugs is that they have significant toxic side effects.
2. Line 64. A clinical vaccine is not available yet but there are one for animal use.
3. Line 77. There are different types of leishmaniasis and models have been developed for these different types e.g. cutaneous and visceral and most in vivo studies use small animal models. Even in cutaneous leishmaniasis it is well know that different species exhibit different types and have different immunological controls e.g. L. major and L. mexicana, and that sex has an important impact on diseeae outcome. Animal studies have identified key genes associated with susceptibility to disease and shown that the same genes are involved in clinical susceptibility e.g. SLC11A1 (formerly NRAMP1, Blackwell et al. 2001, cellular Microbiology 3: 773). So I think this statement is misleading as I think researchers use different animal models for a reason, so I think this part of the introduction needs to be amended to take this into account. Understanding of animal welfare has developed over time. For example, ARRIVE (Animal Research: Reporting of In Vivo Experiments) which provide a checklist of recommendations to improve the reporting of research involving animals guidelines, were developed before 2010, the updated ARRIVE 2.0 guidelines were published in 2020 (check the statement on page 318).
4. Could the authors explain what criteria were used to resolve discrepant results (line 112).
5. The authors are making a novel statement by stating that ‘BALB/c mice are preferred over other strains as they present a high susceptibility phenotype’ (line 239). This has been known by Leishmania researchers for many years.
6. The authors recommend the sue of ‘standardised quantitative PCR methods for VL and CL’ – this means that parasite burdens can only be assessed once in animals in experiments, unless the authors know a way to take biopsy samples from animals with VL and CL. Does this method pick up dead parasites too? What about in vivo imaging using luciferase/fluorescent parasites? This can allow parasite burdens to be determined earlier and over multiple time points in the same animal.
7. Local ethical approval and national approval is required for animal experiments and this means that researchers have to say what adverse effects are likely to occur in animals and what steps are being taken to mitigate adverse effects. So that statement that `The study protocol must always be ethically reviewed by a competent committee’ is misleading and should be amended as it does not take into work researchers and ethical review bodies do before any in vivo studies are completed. I think what is required is clear guidelines on what information should be provided in a manuscript and all journals should ask for the same information and reviewers need to be trained in checking for this information.
Author Response
COMMENTS REVIEWER 2
This is an interesting manuscript and it reviews an important area. I have the following suggestions the authors may like to consider.
- Line 62. I suggest the authors amend the sentence to remove the word dangerous and this is misleading. Clinicians do not administer dangerous levels of drugs – they treat with a prescribed dosage based on the person’s weight and other factors. The problem with many antileishmanial drugs is that they have significant toxic side effects.
R: We have modified the text accordingly
- Line 64. A clinical vaccine is not available yet but there are one for animal use.
R: We have now mentioned that a vaccine for human disease is not available yet.
- Line 77. There are different types of leishmaniasis and models have been developed for these different types e.g. cutaneous and visceral and most in vivostudies use small animal models. Even in cutaneous leishmaniasis it is well know that different species exhibit different types and have different immunological controls e.g. major and L. mexicana, and that sex has an important impact on diseeae outcome. Animal studies have identified key genes associated with susceptibility to disease and shown that the same genes are involved in clinical susceptibility e.g. SLC11A1 (formerly NRAMP1, Blackwell et al. 2001, cellular Microbiology 3: 773). So I think this statement is misleading as I think researchers use different animal models for a reason, so I think this part of the introduction needs to be amended to take this into account. Understanding of animal welfare has developed over time. For example, ARRIVE (Animal Research: Reporting of In Vivo Experiments) which provide a checklist of recommendations to improve the reporting of research involving animals guidelines, were developed before 2010, the updated ARRIVE 2.0 guidelines were published in 2020 (check the statement on page 318).
R: Indeed different models can be used in particular to study immunological responses. This (immunology) is however not the scope of the review, which is focussed on studies that determine the efficacy of (experimental) drugs for leishmaniasis. This is mentioned in the introduction and we do not mention immunology studies.
We have now mentioned that the ARRIVE guidelines have been updated in 2020.
- Could the authors explain what criteria were used to resolve discrepant results (line 112).
R: See line 114: Discrepant results were resolved by discussion until a unanimous decision was reached.
- The authors are making a novel statement by stating that ‘BALB/c mice are preferred over other strains as they present a high susceptibility phenotype’ (line 239). This has been known by Leishmania researchers for many years.
R: We cannot find this comment in line 239. Possibly the reviewer is addressing the statement made in the discussion (lines 340-345). We do not intend to make a novel stamen here, indeed this is known and also we have provided a reference for this. To make it more clear we have now modified the sentence as follows: BALB/c mice are preferred over other strains as they are known to present a high susceptibility phenotype.
- The authors recommend the sue of ‘standardised quantitative PCR methods for VL and CL’ – this means that parasite burdens can only be assessed once in animals in experiments, unless the authors know a way to take biopsy samples from animals with VL and CL. Does this method pick up dead parasites too? What about in vivoimaging using luciferase/fluorescent parasites? This can allow parasite burdens to be determined earlier and over multiple time points in the same animal.
R: We propose to use qPCR as this method is also being used in clinical studies in humans and seem to be provide a good reflection of the actual parasite burden (see for example: Verrest L, et al. Blood Parasite Load as an Early Marker to Predict Treatment Response in Visceral Leishmaniasis in Eastern Africa. Clin Infect Dis. 2021 Sep 7;73(5):775-782. doi: 10.1093/cid/ciab124). Indeed taking multiple biopsy samples might be difficult but certainly not impossible, in particular in the case of CL where the lesions can be easily accessed. We propose this technology as it is relative easy to implement in the laboratory and can be standardised between laboratories. The suggestion of using in vivo imaging using luciferase/fluorescent parasites is an interesting one, but technical much more difficult to implement. As a first step towards harmonization of animal experiments to study experimental leishmaniasis drugs we therefore propose first the use of qPCR. We have now added in the discussion the suggestion that also in vivo imaging using luciferase/fluorescent parasites could be used as a more advanced method, in particular as it would allow to determine parasite burden over multiple time points in the same animal. See discussion: To allow determination of Leishmania parasite burden over multiple time points in the same animal, the use of in vivo imaging using luciferase/fluorescent parasites could be used as a more advanced method. However, this technology might be technically more demanding, requires the use of genetically modified Leishmania strains and extensive laboratory infrastructure.
- Local ethical approval and national approval is required for animal experiments and this means that researchers have to say what adverse effects are likely to occur in animals and what steps are being taken to mitigate adverse effects. So that statement that `The study protocol must always be ethically reviewed by a competent committee’ is misleading and should be amended as it does not take into work researchers and ethical review bodies do before any in vivostudies are completed. I think what is required is clear guidelines on what information should be provided in a manuscript and all journals should ask for the same information and reviewers need to be trained in checking for this information.
R: we fully agree with the reviewer that ethical approval is required. However, many papers were not included in this SR as they failed to report on this. Also welfare aspects and consideration of 3Rs are poorly reported/considered. Thus ethical review must be reported to our opinion and this is clearly not always done. Also guidelines (like ARRIVE) need to be adhered to and journals/reviewers should take these into account. This is all addressed in the first paragraph of the discussion.
Round 2
Reviewer 2 Report
I would like to thank the authors for looking at the suggested changes. I have a few more the authors may like to consider.
1. Title: ‘Animal models for drugs discovery for leishmaniasis: a systematic review’ this does not see to read well. A better one could be ‘Leishmania animal models used in drug discovery: a systematic review’
2. I still do not understand the criteria used to for ‘discrepant results - until a unanimous decision was reached’ means. Can the authors give an example to show what was considered?
3. Line 341 – ‘22 ± 2 gram’ I think the symbol for ± is incorrect.
Other than the title - the manuscript looks fine to me.
Author Response
We thank the reviewer again for his/her valuable input. With respect to the comments we want to respond as follows:
1) Title: we much like the proposed title and we have changed it according to the suggestion of the reviewer.
2) An example of such a discussion could be on the number of animals used in an experiment. For example. reviewer 1 has rejected the manuscript, because the number of animals was not stipulated in the materials and methods section. Reviewer 2 does not agree, because he has observed that you can extract the numbe rof animals used from let say figure 2. During the discussion reviewer 2 will show reviewer 1 where to find this information and they will come to a mutual agreement. I hope that this clarifies this point. Perhaps to add that we did not had many disagreements, mainly due to the fact that out inclusion and exclusion were well described.
3) Thank you for this observation; we have modified it.